# BROS: A Pre-trained Language Model for Understanding Texts in Document

## Abstract

Understanding document from their visual snapshots is an emerging and challenging problem that requires both advanced computer vision and NLP methods. Although the recent advance in OCR enables the accurate extraction of text segments, it is still challenging to extract key information from documents due to the diversity of layouts. To compensate for the difficulties, this paper introduces a pre-trained language model, *BERT Relying On Spatiality (BROS)*, that represents and understands the semantics of spatially distributed texts. Different from previous pre-training methods on 1D text, BROS is pre-trained on large-scale semi-structured documents with a novel area-masking strategy while efficiently including the spatial layout information of input documents. Also, to generate structured outputs in various document understanding tasks, BROS utilizes a powerful graph-based decoder that can capture the relation between text segments. BROS achieves state-of-the-art results on four benchmark tasks: FUNSD, SROIE*, CORD, and SciTSR. *Our experimental settings and implementation codes will be publicly available.*

## 1    Introduction

Document intelligence (DI)[1], which understands industrial documents from their visual appearance, is a critical application of AI in business. One of the important challenges of DI is a key information extraction task (KIE) (Huang et al., 2019; Jaume et al., 2019; Park et al., 2019) that extracts structured information from documents such as financial reports, invoices, business emails, insurance quotes, and many others. KIE requires a multi-disciplinary perspective spanning from computer vision for extracting text from document images to natural language processing for parsing key information from the identified texts.

Optical character recognition (OCR) is a key component to extract texts in document images. As OCR provides a set of text blocks consisting of a text and its location, key information in documents can be represented as a single or a sequence of the text blocks (Schuster et al., 2013; Qian et al., 2019; Hwang et al., 2019; 2020). Although OCR alleviates the burden of processing images, understanding semantic relations between text blocks on diverse layouts remains a challenging problem.

To solve this problem, existing works use a pre-trained language model to utilize its effective representation of text. Hwang et al. (2019) fine-tunes BERT by regarding KIE tasks as sequence tagging problems. Denk & Reisswig (2019) uses BERT to incorporate textual information into image pixels during their image segmentation tasks. However, since BERT is designed for text sequences, they artificially convert text blocks distributed in two dimensions into a single text sequence losing spatial layout information. Recently, Xu et al. (2020) proposes LayoutLM pre-trained on large-scale documents by utilizing spatial information of text blocks. They show the effectiveness of the pre-training approach by achieving high performance on several downstream tasks. Despite this success, LayoutLM has three limitations. First, LayoutLM embeds x- and y-axis individually using trainable parameters like the position embedding of BERT, ignoring the gap between positions in a sequence and 2D space. Second, its pre-training method is essentially identical to BERT that does not explicitly consider spatial relations between text blocks. Finally, in its downstream tasks, LayoutLM only conducts sequential tagging tasks (e.g. BIO tagging) that require serialization of text blocks.

---

[1] https://sites.google.com/view/di2019

These limitations indicate that LayoutLM fails not only to fully utilize spatial information but also to address KIE problems in practical scenarios when a serialization of text blocks is difficult.

This paper introduces an advanced language model, BROS, pre-trained on large-scale documents, and provides a new guideline for KIE tasks. Specifically, to address the three limitations mentioned above, BROS combines three proposed methods: (1) a 2D positional encoding method that can represent the continuous property of 2D space, (2) a novel area-masking pre-training strategy that performs masked language modeling on 2D, and (3) a combination with a graph-based decoder for solving KIE tasks. We evaluated BROS on four public KIE datasets: FUNSD (form-like documents), SROIE* (receipts), CORD (receipts), and SciTSR (table structures) and observed that BROS achieved the best results on all tasks. Also, to address KIE problem under a more realistic setting we removed the order information between text blocks from the four benchmark datasets. BROS still shows the best performance on these modified datasets. Further ablation studies provide how each component contributes to the final performances of BROS.

## 2 RELATED WORK

### 2.1 PRE-TRAINED LANGUAGE MODELS

BERT (Devlin et al., 2019) is a pre-trained language model using Transformer (Vaswani et al., 2017) that shows superior performance on various NLP tasks. The main strategy to train BERT is a masked language model (MLM) that masks and estimates randomly selected tokens to learn the semantics of language from large-scale corpora. Many variants of BERT have been introduced to learn transferable knowledge by modifying the pre-training strategy. XLNet (Yang et al., 2019) permutes tokens during the pre-training phase to reduce a discrepancy from the fine-tuning phase. XLNet also utilizes relative position encoding to handle long texts. StructBERT (Wang et al., 2020) shuffles tokens in text spans and adds sentence prediction tasks for recovering the order of words or sentences. SpanBERT (Joshi et al., 2020) masks span of tokens to extract better representation for span selection tasks such as question answering and co-reference resolution. ELECTRA (Clark et al., 2020) is trained to distinguish real and fake input tokens generated by another network for sample-efficient pre-training.

Inspired by these previous works, BROS utilizes a new pre-training strategy that can capture complex spatial dependencies between text blocks distributed on two dimensions. Note that LayoutLM is the first pre-trained language model on spatial text blocks but they still employs the original MLM of BERT.

### 2.2 KEY INFORMATION EXTRACTION FROM DOCUMENTS

Most of the existing approaches utilize a serializer to identify the text order of key information. POT (Hwang et al., 2019) applies BERT on serialized text blocks and extracts key contexts via a BIO tagging approach. CharGrid (Katti et al., 2018) and BERTGrid (Denk & Reisswig, 2019) map text blocks upon a grid space, identify the region of key information, and extract key contexts in the pre-determined order. Liu et al. (2019), Yu et al. (2020), and Qian et al. (2019) utilize graph convolutional networks to model dependencies between text blocks but their decoder that performs BIO tagging relies on a serialization. LayoutLM (Xu et al., 2020) is pre-trained on large-scale documents with spatial information of text blocks, but it also conducts BIO tagging for their downstream tasks. However, using a serializer and relying on the identified sequence has two limitations. First, the information represented in two dimensional layout can be lost by improper serialization. Second, there may even be no correct serialization order.

A natural way to model key contexts from text blocks is a graph-based formulation that identifies all relationships between text blocks. SPADE (Hwang et al., 2020) proposes a graph-based decoder to extract key contexts from identified connectivity between text blocks without any serialization. Specifically, they utilize BERT without its sequential position embeddings and train the model while fine-tuning BERT. However, their performance is limited by the amount of data as all relations have to be learned from the beginning at the fine-tuning stage. To fully utilize the graph-based decoder, BROS is pre-trained on a large number of documents and is combined with the SPADE decoder to determine key contexts from text blocks.

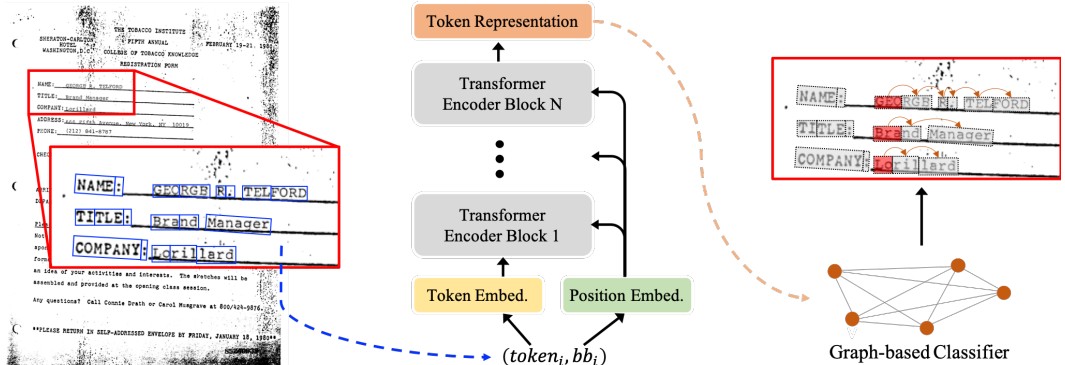

Figure 1: Overview of BROS for downstream KIE tasks. Tokens in a document are feed into BROS with their bounding box information. After pre-trained with large-scale unlabeled documents, BROS is fine-tuned with a graph-based decoder for document KIE tasks that have small labeled documents.

## 3 BERT RELYING ON SPATIALITY (BROS)

The main structure of BROS follows BERT, but there are three novel differences: (1) a spatial encoding metric that reflects the continuous property of 2D space, (2) a pre-training objective designed for text blocks on 2D space, and (3) a guideline for designing downstream models based on a graph-based formulation. Figure 1 shows visual description of BROS for document KIE tasks.

### 3.1 ENCODING SPATIAL INFORMATION INTO BERT

#### 3.1.1 REPRESENTATION OF A TEXT BLOCK LOCATION

The way to represent spatial information of text blocks is important to encode information from layouts. We utilize sinusoidal functions to encode continuous values of x- and y-axis, and merge them through a linear transformation to represent a point upon 2D space.

For formal description, we use $\boldsymbol{p} = (x, y)$ to denote a point on 2D space and $\mathbf{f}^{\text{sinu}} : \mathbb{R} \to \mathbb{R}^{D^{\text{s}}}$ to represent a sinusoidal function. $D^{\text{s}}$ is the dimensions of sinusoid embedding. BROS encodes a 2D point by applying the sinusoidal function to x- and y-axis and concatenating them as $\bar{\boldsymbol{p}} = [\mathbf{f}^{\text{sinu}}(x) \oplus \mathbf{f}^{\text{sinu}}(y)]$. The $\oplus$ symbol indicates concatenation. The bounding box of a text block, $bb_i$, consists of four vertices, such as $\boldsymbol{p}_i^{\text{tl}}, \boldsymbol{p}_i^{\text{tr}}, \boldsymbol{p}_i^{\text{br}}$, and $\boldsymbol{p}_i^{\text{bl}}$ that indicate top-left, top-right, bottom-right, and bottom-left points, respectively. The four points are converted into vectors such as $\bar{\boldsymbol{p}}_i^{\text{tl}}, \bar{\boldsymbol{p}}_i^{\text{tr}}, \bar{\boldsymbol{p}}_i^{\text{br}}$, and $\bar{\boldsymbol{p}}_i^{\text{bl}}$ with $\mathbf{f}^{\text{sinu}}$. Finally, to represent a spatial embedding, $\overline{bb}_i$, BROS combines four identified vectors through a linear transformation,

$$\overline{bb}_i = \boldsymbol{W}^{\text{tl}} \bar{\boldsymbol{p}}_i^{\text{tl}} + \boldsymbol{W}^{\text{tr}} \bar{\boldsymbol{p}}_i^{\text{tr}} + \boldsymbol{W}^{\text{br}} \bar{\boldsymbol{p}}_i^{\text{br}} + \boldsymbol{W}^{\text{bl}} \bar{\boldsymbol{p}}_i^{\text{bl}}, \tag{1}$$

where $\boldsymbol{W}^{\text{tl}}, \boldsymbol{W}^{\text{tr}}, \boldsymbol{W}^{\text{br}}, \boldsymbol{W}^{\text{bl}} \in \mathbb{R}^{H \times 2D^{\text{s}}}$ are linear transition metrics and $H$ is a hidden size of BERT. The periodic property of the sinusoidal function can encode continuous 2D coordinates more natural than using point-specific embedding used in BERT and LayoutLM. In addition, by learning the linear transition parameters, BROS provides an effective representation of a bounding box.

#### 3.1.2 ENCODING SPATIAL REPRESENTATION

Position encoding methods affect how models utilize the position information. In BERT, position embedding is tied with the token through a point-wise summation. However, 2D spatial information is richer than 1D sequence due to the their continuous property and the high dimensionality. Moreover, text blocks can be placed over various locations on documents without significant changes in its semantic meaning. For example, locations of page numbers differ over multiple document snapshots even though they are captured from a single document. Therefore, more advanced approach is required to maximally include spatial information during encoding beyond the simple summation approach used in BERT. In BROS, the spatial information is directly encoded during the contextualization of text blocks. Specifically, BROS calculates an attention logit combining both semantic and

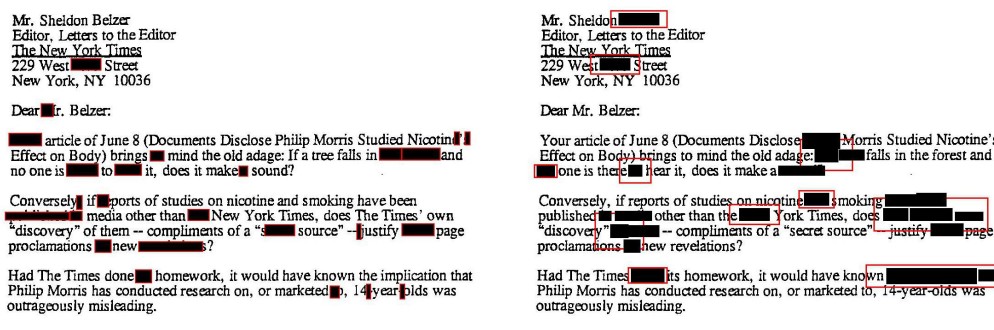

(a) Random *token* selection and *token* masking  (b) Random *area* selection and *block* masking

Figure 2: Comparison of (a) conventional token masking and (b) area masking. Although there can be multiple tokens in a single text block, the token masking flats all tokens of a document, selects a token randomly (red), and masks it directly (black). In contrast, the area masking selects a text block, identifies an area by expanding the block area with an exponential distribution (red), masks tokens in all text blocks of which center is aligned in the identified area (black). In both, 15% of tokens are masked.

spatial features. The former is the same as the original attention mechanism in Transformer (Vaswani et al., 2017), but the latter is a new component identifying the importance of the target location when the source context and location are given. Our proposed attention logit is formulated as follows,

$$A_{i,j} = (\boldsymbol{W}^{\mathrm{q}}\boldsymbol{t}_i)^\top(\boldsymbol{W}^{\mathrm{k}}\boldsymbol{t}_j) + (\boldsymbol{W}^{\mathrm{q}}\boldsymbol{t}_i \odot \boldsymbol{W}^{\mathrm{sq|q}}\overline{\boldsymbol{bb}}_i)^\top(\boldsymbol{W}^{\mathrm{sk|q}}\overline{\boldsymbol{bb}}_j) + (\boldsymbol{W}^{\mathrm{sq}}\overline{\boldsymbol{bb}}_i)^\top(\boldsymbol{W}^{\mathrm{sk}}\overline{\boldsymbol{bb}}_j), \quad (2)$$

where $\boldsymbol{t}_i$ and $\boldsymbol{t}_j$ are context representations for $i^{\mathrm{th}}$ and $j^{\mathrm{th}}$ tokens and $\boldsymbol{W}^{\mathrm{q}}$, $\boldsymbol{W}^{\mathrm{k}}$, $\boldsymbol{W}^{\mathrm{sq|q}}$, $\boldsymbol{W}^{\mathrm{sk|q}}$, $\boldsymbol{W}^{\mathrm{sq}}$, $\boldsymbol{W}^{\mathrm{sk}}$ are linear transition matrices. The $\odot$ symbol indicates Hadamard product. The first term indicates an attention logit from contextual representations and the third term is from spatial representation. The second term is designed to model the spatial dependency given the source semantic representation, $\boldsymbol{t}_i$. The second and third terms are independently calculated at each layer because spatial dependencies might differ over layers.

## 3.2 PRE-TRAINING OBJECTIVE: AREA-MASKED LANGUAGE MODEL

Pre-training diverse layouts from unlabeled documents is a key factor for document understanding tasks. To learn effective spatial representation including relationships between text blocks, we propose a novel pre-training objective. Inspired by SpanBERT (Joshi et al., 2020), we expand spans of a 1D sequence to consecutive text blocks in 2D space. Specifically, we select a few regions in a document layout, mask all tokens of text blocks in the selected regions, and estimate the masked tokens. The rules for masking tokens in area-masked language model are as the following procedure.

(a) Select a text block randomly and get the top-left and bottom-right points ($\boldsymbol{p}^{\mathrm{tl}}$ and $\boldsymbol{p}^{\mathrm{br}}$) of the block.
(b) Identify the width, height, and center of the block as $(w, h) = |\boldsymbol{p}^{\mathrm{tl}} - \boldsymbol{p}^{\mathrm{br}}|$ and $\mathbf{c} = (\boldsymbol{p}^{\mathrm{tl}} + \boldsymbol{p}^{\mathrm{br}})/2$.
(c) Expand the width and height as $(\hat{w}, \hat{h}) = l * (w, h)$ where $l \sim \exp(\lambda)$ and $\lambda$ is a distribution parameter.
(d) Identify rectangular masking area of which top-left and bottom-right are $\hat{\boldsymbol{p}}^{\mathrm{tl}} = \boldsymbol{p}^{\mathrm{tl}} - (\hat{w}, \hat{h})$, and $\hat{\boldsymbol{p}}^{\mathrm{br}} = \boldsymbol{p}^{\mathrm{br}} + (\hat{w}, \hat{h})$, respectively.
(e) Mask all tokens of text blocks whose centers are allocated in the area.
(f) Repeat (a)–(e) until 15% of tokens are masked.

The rationale behind using exponential distribution is to convert the geometric distribution used in SpanBERT for a discrete domain into distribution for a continuous domain. Thus, we set $\lambda = -\ln(1 - p)$ where $p = 0.2$ used in SpanBERT. In addition, we truncated exponential distribution with 1 to prevent an infinity multiplier covering all space of the document. It should be noted that the masking area is expanded from a randomly selected text block since the area should be related to the text sizes and locations to represent text spans in 2D space. Figure 2 compares token- and area-masking on text blocks.

Finally, the loss function for the area-masked language model is formed as;

$$L_{\text{AMLM}} = - \sum_{\hat{x} \in A(\mathbf{x})} \log p(\hat{x}|\mathbf{x}_{\backslash A(\mathbf{x})}), \tag{3}$$

where $\mathbf{x}$, $A(\mathbf{x})$, and $\mathbf{x}_{\backslash A(\mathbf{x})}$ denote tokens in given image, masked tokens of which text block is located in masking area, and the rest tokens, respectively. Similar to BERT (Devlin et al., 2019), the masked tokens are replaced by [MASK] token 80% of the time, a random token 10% of the time, or an unchanged token 10% of the time.

### 3.3 SPATIAL DEPENDENCY PARSERS FOR DOWNSTREAM TASKS

Key information in a document (e.g. store address in a receipt) is represented as sub-sequences of text blocks. Although BIO tagging has been used to extract the sub-sequences from a text sequence, it cannot represent key texts in a document without the optimal order of text blocks. Therefore, BIO tagging cannot be applied when the optimal order is not available which often can appear in a practical scenario.

To deal with the issue, BROS utilizes a decoder of SPADE (Hwang et al., 2020) that can infer a sequence of text blocks by employing a graph-based formulation. BROS supports two downstream tasks: (1) an entity extraction (EE) task and (2) an entity linking (EL) task. The EE identifies a sequence of text blocks for key information (e.g. extract address texts in a receipt) and the EL determines relations between target texts when target text blocks are known (e.g. identify key and value text pairs).

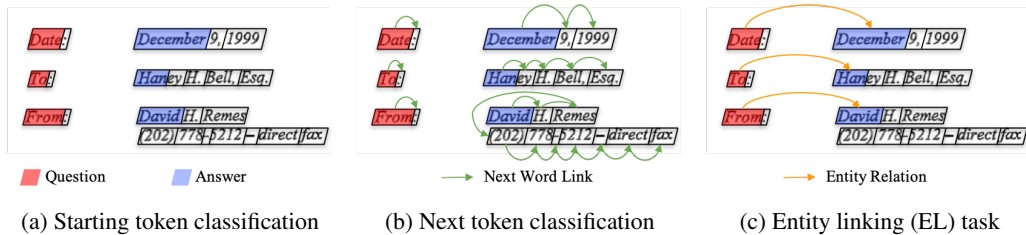

(a) Starting token classification    (b) Next token classification    (c) Entity linking (EL) task

Figure 3: Visual descriptions of BROS downstream tasks. For EE tasks, BROS combines two sub-tasks such as (a) and (b). For EL task, BROS links the first tokens of the entities.

For EE tasks, BROS divides the problem into two sub-tasks: starting token classification (Figure 3, a) and subsequent token classification (Figure 3, b). Let $\tilde{\boldsymbol{t}}_i \in \mathbb{R}^H$ denote the $i^{\text{th}}$ token representation from the last Transformer layer of BROS. The starting token classification conducts a token-level tagging determining whether a token is a starting token of target information as follows,

$$\mathbf{f}_{\text{stc}}(\tilde{\boldsymbol{t}}_i) = \text{softmax}(\boldsymbol{W}^{\text{stc}}\tilde{\boldsymbol{t}}_i) \in \mathbb{R}^{C+1}, \tag{4}$$

where $\boldsymbol{W}^{\text{stc}} \in \mathbb{R}^{(C+1) \times H}$ is a linear transition matrix and $C$ indicates the number of target classes. Here, the extra +1 dimension is considered to indicate non-starting tokens.

The subsequent token classification is conducted by utilizing pair-wise token representations as follows,

$$\mathbf{f}_{\text{ntc}}(\tilde{\boldsymbol{t}}_i) = \text{softmax}((\boldsymbol{W}^{\text{ntc-s}}\tilde{\boldsymbol{t}}_i)^{\top}(\boldsymbol{t}^{ntc} \oplus \boldsymbol{W}^{\text{ntc-t}}\tilde{\boldsymbol{t}}_1 \oplus \cdots \oplus \boldsymbol{W}^{\text{ntc-t}}\tilde{\boldsymbol{t}}_N))^{\top} \in \mathbb{R}^{N+1}, \tag{5}$$

where $\boldsymbol{W}^{\text{ntc-s}}, \boldsymbol{W}^{\text{ntc-t}} \in \mathbb{R}^{H^{\text{ntc}} \times H}$ are linear transition matrices, $H^{\text{ntc}}$ is a hidden feature dimension for the next token classification decoder and $N$ is the maximum number of tokens. Here, $\boldsymbol{t}^{\text{ntc}} \in \mathbb{R}^{H^{\text{ntc}}}$ is a model parameter to classify tokens which do not have a next token or are not related to any class. It has a similar role with an end-of-sequence token, [EOS], in NLP. By solving these two sub-tasks, BROS can identify a sequence of text blocks by finding first tokens and connecting subsequent tokens.

For EL tasks, BROS conducts a binary classification for all possible pairs of tokens (Figure 3, c) as follows,

$$\mathbf{f}_{\text{rel}}(\tilde{\boldsymbol{t}}_i, \tilde{\boldsymbol{t}}_j) = \text{sigmoid}((\boldsymbol{W}^{\text{rel-s}}\tilde{\boldsymbol{t}}_i)^{\top}(\boldsymbol{W}^{\text{rel-t}}\tilde{\boldsymbol{t}}_j)), \tag{6}$$

where $\boldsymbol{W}^{\text{rel-s}}$, $\boldsymbol{W}^{\text{rel-t}} \in \mathbb{R}^{H^{\text{rel}} \times H}$ are linear transition matrices and $H^{\text{rel}}$ is a hidden feature dimension. Compared to the subsequent token classification, a single token can hold multiple relations with other tokens to represent hierarchical structures of document layouts. For more detail about this graph-based formulation, see Appendix E.

## 4    KEY INFORMATION EXTRACTION TASKS

Here, we describe three EE tasks and three EL tasks from four KIE benchmark datasets.

- Form Understanding in Noisy Scanned Documents (FUNSD) (Jaume et al., 2019) is a set of documents with various forms. The dataset consists of 149 training and 50 testing examples. FUNSD has both EE and EL tasks. In the EE task, there are three semantic entities: Header, Question, and Answer. In the EL task, the semantic hierarchies are represented as relations between text blocks like header-question and question-answer pairs.
- SROIE* is a variant of Task 3 of "Scanned Receipts OCR and Information Extraction" (SROIE)[2] that consists of a set of store receipts. In the original SROIE task, semantic contents (Company, Date, Address, and Total price) are generated without explicit connection to the text blocks. To convert SROIE into a EE task, we developed SROIE* by matching ground truth contents with text blocks. We also split the original training set into 526 training and 100 testing examples because the ground truths are not given in the original test set. SROIE* will be publicly available.
- Consolidated Receipt Dataset (CORD) (Park et al., 2019) is a set of store receipts with 800 training, 100 validation, and 100 testing examples. CORD consists of both EE and EL tasks. In the EE task, there are 30 semantic entities including menu name, menu price, and so on. In the EL task, the semantic entities are linked according to their layout structure. For example, menu name entities are linked to menu id, menu count, and menu price.
- Complicated Table Structure Recognition (SciTSR) (Chi et al., 2019) is a EL task that connects cells in a table to recognize the table structure. There are two types of relations: vertical and horizontal connection between cells. The dataset consists of 12,000 training images and 3,000 test images.

Although, these four datasets provide test beds for the EE and EL tasks, they represent the subset of real problems as the optimal order of text blocks is given. In real service, user can submit documents with a complex layout where the serialization of input texts are non-trivial. FUNSD provides the optimal orders of text blocks related to target classes in both training and testing examples. In SROIE*, CORD, and SciTSR, the text blocks are serialized in reading orders.

To reveal the serialization problem in the EE and EL tasks, we randomly permuted text blocks of the datasets to remove their order information. We denote the permuted datasets as p-FUNSD, p-SROIE*, p-CORD, and p-SciTSR and compare all models on them. For fair comparisons, we will open the permuted datasets.

## 5    EXPERIMENTS

### 5.1    EXPERIMENT SETTINGS

For pre-training, IIT-CDIP Test Collection 1.0[3] (Lewis et al., 2006), which consists of approximatley 11M document images, is used but 400K RVL-CDIP dataset[4] (Harley et al., 2015) is excluded following LayoutLM. In-house OCR engine was applied to obtain text blocks from unlabeled document images.

The main Transformer structure of BROS is the same as BERT. By following BERT$_{\text{BASE}}$, the hidden size, the number of self-attention heads, the feed-forward size, and the number of Transformer layers set to 768, 12, 3072, and 12, respectively. The same pre-training setting with LayoutLM is used for a fair comparison.

---

[2]https://rrc.cvc.uab.es/?ch=13

[3]https://ir.nist.gov/cdip/

[4]https://www.cs.cmu.edu/ aharley/rvl-cdip/

| Dataset | Model | Entity Extraction | | | Entity Linking | | |
|---|---|---|---|---|---|---|---|
| | | P | R | F1 | P | R | F1 |
| FUNSD | BERT (Xu et al., 2020) | 54.69 | 61.70 | 60.26 | 30.89 | 25.03 | 27.65 |
| | LayoutLM (Xu et al., 2020) | 75.97 | 81.55 | 78.66 | - | - | - |
| | LayoutLM$^\dagger$ | 76.12 | 81.88 | 78.89 | 47.80 | 48.21 | 48.00 |
| | BROS | **80.56** | **81.88** | **81.21** | **64.70** | **70.83** | **67.63** |
| SROIE* | BERT | 92.90 | 94.47 | 93.67 | | | |
| | LayoutLM$^\dagger$ | 94.31 | 95.78 | 95.04 | | - | |
| | BROS | **94.93** | **96.03** | **95.48** | | | |
| CORD | BERT | 93.08 | 93.18 | 93.13 | 93.03 | 92.62 | 92.83 |
| | LayoutLM$^\dagger$ | 95.03 | 94.58 | 94.80 | 93.58 | 92.72 | 93.15 |
| | BROS | **95.58** | **95.14** | **95.36** | **94.37** | **92.80** | **93.58** |
| SciTSR | Tabby (Shigarov et al., 2016) | | | | 92.6 | 92.0 | 92.1 |
| | DeepDeSRT (Schreiber et al., 2017) | | | | 90.6 | 88.7 | 89.0 |
| | GraphTSR (Chi et al., 2019) | | - | | 95.9 | 94.8 | 95.3 |
| | BERT | | | | 87.61 | 85.92 | 86.76 |
| | LayoutLM$^\dagger$ | | | | 98.76 | 99.44 | 99.09 |
| | BROS | | | | **99.05** | **99.47** | **99.26** |

Table 1: Performance comparisons on three EE and three EL tasks with the *optimal order information* of text blocks. P, R, and F1 indicate precision, recall, and F1 scores, respectively. For EL tasks connecting entities, SPADE decoder is applied for all pre-trained models such as BERT, LayoutLM, and BROS. The scores are the average of the results with 5 different random seeds.

BROS is trained by using AdamW optimizer (Loshchilov & Hutter, 2019) with a learning rate of 5e-5 with linear decay. First 10% of the total epochs are used for a warm-up. We initialized weights of BROS with those of BERT$_{\text{BASE}}$ and trained BROS on IIT-CDIP for 2 epochs with 80 of batch size, following LayoutLM. The pre-training takes 64 hours on 8 NVIDIA Tesla V100s with Distributed Data Parallel (DDP).

During fine-tuning, the learning rate is set to 5e-5. The batch size is set to 16 for all tasks. The number of training epochs or steps is as follows: 100 epochs for FUNSD, 1K steps for SROIE* and CORD, and 7.5 epochs for SciTSR. The hidden feature dimensions, $H^{\text{ntc}}$ and $H^{\text{rel}}$, of the SPADE decoder are set to 128 for FUNSD, 64 for SROIE*, and 256 for CORD and SciTSR.

Although the authors of LayoutLM published their codes on GitHub[5], the data and script file used for pre-training are not included. For a fair comparison, we made our own implementation, which we refer to LayoutLM$^\dagger$, on the same pre-training data and script file used for BROS pre-training. We verified LayoutLM$^\dagger$ by comparing its performances on FUNSD from the reported scores in (Xu et al., 2020). See Appendix A for more information.

## 5.2 EXPERIMENTAL RESULTS WITH OPTIMAL ORDER INFORMATION

Table 1 shows the results on four KIE datasets with given optimal order information of text blocks. For EL tasks, we applied SPADE decoders to all pre-trained models such as BERT, LayoutLM, and BROS. In all tasks, we observed that BERT shows lower scores than LayoutLM and BROS presumably due to the loss of spatial information. BROS achieves the highest scores showing the effectiveness of our approach. Specifically, in FUNSD, BROS shows the state-of-the-art performances with a large margins of 2.32pp in the EE task and 19.63pp in the EL task. Moreover, it should be noted that BROS achieves higher F1 score than one of the LayoutLM variants, which utilizes visual features (81.21 > 79.27 (Xu et al., 2020)). In SROIE* and CORD, BROS also shows the best performances over all the EE and EL tasks. In SciTSR, LayoutLM and BROS show the importance of pre-training by exceeding other baselines with large margins which are trained by using either only spatial information of cells (Tabby and DeepDeSRT) or without pre-training spatial texts

---
[5]https://github.com/microsoft/unilm/tree/master/layoutlm

(GraphTSR). These results prove that BROS captures better representations of text blocks for KIE downstream tasks.

## 5.3 EXPERIMENTAL RESULTS WITHOUT OPTIMAL ORDER INFORMATION

It is an another challenging problem to arrange text blocks in the order that humans can understand (Li et al., 2020). Although most commercial OCR products provide an order of OCR text blocks, they are unable to reconcile the structural formatting of the texts precisely (See Appendix B). Therefore, the experiments in Section 5.2 cannot fully represent real KIE problems because they assume the optimal order of text blocks is given. To reveal the challenge, we removed the order information in all datasets by permuting the order of text blocks as mentioned in Section 4 and investigated how BERT, LayoutLM, and BROS work without the order information. We utilized a SPADE decoder for all models because BIO tagging on these permuted dataset cannot extract a sequence of text blocks in a correct order.

| Dataset | Model | Entity Extraction | | | Entity Linking | | |
|---|---|---|---|---|---|---|---|
| | | P | R | F1 | P | R | F1 |
| p-FUNSD | BERT | 21.17 | 16.99 | 18.85 | 14.13 | 7.26 | 9.59 |
| | LayoutLM[†] | 34.98 | 32.55 | 33.72 | 28.73 | 23.67 | 25.95 |
| | BROS | **74.84** | **75.50** | **75.14** | **63.94** | **65.53** | **64.73** |
| p-SROIE | BERT | 40.65 | 38.84 | 39.73 | | | |
| | LayoutLM[†] | 67.42 | 66.48 | 66.94 | | - | |
| | BROS | **81.32** | **81.36** | **81.34** | | | |
| p-CORD | BERT | 59.84 | 59.58 | 59.71 | 31.29 | 25.14 | 27.88 |
| | LayoutLM[†] | 77.49 | 77.27 | 77.38 | 56.11 | 53.67 | 54.86 |
| | BROS | **93.86** | **93.68** | **93.77** | **84.77** | **83.15** | **83.95** |
| p-SciTSR | BERT | | | | 59.42 | 0.89 | 1.75 |
| | LayoutLM[†] | | - | | 95.59 | 99.04 | 97.28 |
| | BROS | | | | **98.90** | **99.32** | **99.11** |

Table 2: Performance comparisons on three EE and three EL tasks without the optimal order information of text blocks. All models utilize the SPADE decoder because BIO tagging cannot be used without the optimal order. The scores are the average of the results with 5 different random seeds.

Table 2 shows the results. Due to the lose of correct orders, BERT shows poor performances over all tasks. By utilizing spatial information of text blocks, LayoutLM[†] shows better performance but it suffers from huge performance degradation compared to the score computed with the optimal order. On the other hand, BROS shows comparable results compared the cases with the optimal order and achieves better performances than BERT and LayoutLM[†].

To systematically investigate how the order information affects the performance of the models, we construct variants of FUNSD by re-ordering text blocks with two sorting methods based on the top-left points. The text blocks of xy-FUNSD are sorted according to x-axis with ascending order of y-axis and those of yx-FUNSD are sorted according to y-axis with ascending order of x-axis.

Table 3 shows performance on p-FUNSD, xy-FUNSD, yx-FUNSD, and the original FUNSD. All models utilize a SPADE decoder for a fair comparison. Interestingly, the performance of LayoutLM[†] is degraded in the order of FUNSD, yx-FUNSD, xy-FUNSD, and p-FUNSD as like the order of the reasonable serialization for text on 2D space. On the other hand, the performance of BROS is relatively consistent. These results show that BROS is applicable to real KIE problems without relying on an additional serialization method.

## 5.4 ABLATION STUDIES

Table 4 provides the result of the ablative experiments computed while changing pre-training strategy, 2D position embedding and encoding methods, and a decoder for downstream tasks. The 2D embedding method represents how to treat spatial information of text blocks and the 2D encoding

| Model | p-FUNSD | xy-FUNSD | yx-FUNSD | FUNSD |
|---|---|---|---|---|
| LayoutLM[†] | 33.72 | 36.61 | 60.50 | 77.30 |
| BROS | 75.14 | 75.24 | 76.26 | 81.21 |

Table 3: Comparison of the EE task performance changes according to sorting methods. p-FUNSD and FUNSD indicate permuted and original datasets, repectively. xy-FUNSD/yx-FUNSD represents datasets in which text blocks are sorted on x/y-axis with the top-left point of the blocks and then sorted on y/x-axis again if holding the same values. For the same reason as in Table 2, all model utilize a SPADE decoder. The scores indicate averages of F1 scores for the EE task and they are repeated five times.

| **MLM** | **F1** | **2D position embedding method** | **F1** |
|---|---|---|---|
| Random tokens (LayoutLM's) | 73.85 | Look-up tables (LayoutLM's) | 67.99 |
| Random areas | 74.44 | Sinusoid & linear | 74.44 |

| **2D position encoding method** | **F1** | **Decoder for downstream tasks** | **F1** |
|---|---|---|---|
| Tied with token embedding (LayoutLM's) | 42.46 | BIOE tagging (LayoutLM's) | 73.16 |
| Untied with token embedding | 74.44 | SPADE decoder | 74.44 |

Table 4: Ablative studies of the proposed changes from BROS's to LayoutLM's on the FUNSD EE task. For each ablation, the other components are set as those of BROS. The number of pre-training data is 512K and the scores are the average of 5 experimental results.

method indicates how to merge the 2D embeddings into BERT. The results show that all modifications improve performance when comparing the methods of LayoutLM. Specifically, 2D position embedding and its encoding methods show huge performance gaps by 6.45pp and 31.98pp, respectively. These results represent a co-modality of our 2D continuous position embedding approach and its untied encoding method.

LayoutLM and BROS are initialized with weights of BERT to utilize powerful knowledge of BERT that learns from large-scale corpora. However, BERT includes its 1D positional embeddings (1D-PE) that might be harmful by making a sequence of text blocks even though there is no order information. To investigate the effectiveness of the 1D-PE, we conduct an additional ablative study. BROS without the 1D-PE shows the same F1 scores on both FUNSD and p-FUNSD (70.07), but BROS with the 1D-PE shows performance degradation when the dataset loses the optimal order information (81.21 on FUNSD $\rightarrow$ 75.14 on p-FUNSD). Nevertheless, BROS with the 1D-PE shows better performances on both datasets. This might be because the 1D-PE preserves the token order in a single text block. Based on this result, we decided to incorporate the 1D-PE in our model.

## 6 CONCLUSION

We present a novel pre-trained language model, BROS, for understanding semi-structured documents. BROS encodes 2D continuous position of text blocks and learns natural language from text blocks with an area-driven training strategy. To extract key contexts from text blocks without the order information, BROS adapts a graph-based decoder that identifies text sequences for EE tasks and layout relationships for EL tasks. In our extensive experiments on three EE and three EL tasks, BROS consistently shows better performances as well as the robustness on perturbed orders of text blocks compared to the existing approaches.

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

## A    REPRODUCING THE LAYOUTLM

As mentioned in the paper, to compare BROS from LayoutLM in diverse experimental settings, we implement LayoutLM in our experimental pipeline. Table 5 compares our implementations from the reported scores in Xu et al. (2020). As can be seen, multiple experiments are conducted according to the number of pre-training data. Our implementation, referred to LayoutLM$^{\dagger}$, shows comparable performances over all settings.

| # Pre-training Data | # Epochs | Model | P | R | F1 |
|---|---|---|---|---|---|
| 500K | 1 | LayoutLM (Xu et al., 2020) | 0.5779 | 0.6955 | 0.6313 |
| | | LayoutLM$^{\dagger}$ | 0.5823 | 0.6935 | 0.6330 |
| 1M | 1 | LayoutLM (Xu et al., 2020) | 0.6156 | 0.7005 | 0.6552 |
| | | LayoutLM$^{\dagger}$ | 0.6142 | 0.7151 | 0.6608 |
| 2M | 1 | LayoutLM (Xu et al., 2020) | 0.6599 | 0.7355 | 0.6957 |
| | | LayoutLM$^{\dagger}$ | 0.6562 | 0.7456 | 0.6980 |
| 11M | 1 | LayoutLM (Xu et al., 2020) | 0.7464 | 0.7815 | 0.7636 |
| | | LayoutLM$^{\dagger}$ | 0.7384 | 0.8022 | 0.7689 |
| | 2 | LayoutLM (Xu et al., 2020) | 0.7597 | 0.8155 | 0.7866 |
| | | LayoutLM$^{\dagger}$ | 0.7612 | 0.8188 | 0.7889 |

Table 5: Sanity checking of LayoutLM$^{\dagger}$ by comparing its performances on FUNSD from the reported scores in Xu et al. (2020).

## B    VISUALIZATION OF SERIALIZED OCR BLOCKS

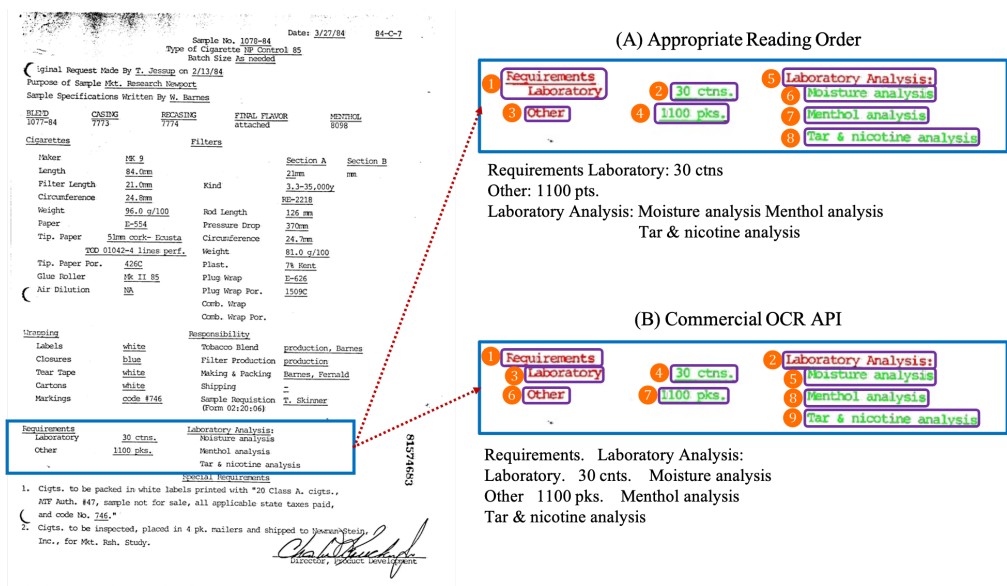

Figure 4: OCR block sequence comparisons between appropriate reading order and commercial OCR (Google Cloud - Vision API) on FUNSD training sample. Red text denotes question, green texts denotes answer, circled number denotes appropriate reading order, order of OCR output each.

With the developments in the field of machine learning, the performance of commercial OCR has improved over the years. However, it is still hard to entrust the ordering of commercial OCR block outputs Li et al. (2020). Figure 4 shows the gap between the comprehensive reading order and outputs of commercial OCR. Specifically, the figure contrasts how the words in the OCR results should be serialized (Figure 4a) but most commercially available OCR technologies are unable to

reconcile the structural formatting of the text – leading to them ordering the words horizontally (Figure 4b). This cursory example illustrates that as advanced as commercial OCR solutions have become, there are still ways to improve and our proposed method is one way in which this can be done.

## C ABLATION STUDIES

Here, we provide more ablation studies on the components proposed in the paper. In the following tables, the number of pre-training data is 512K and the scores (F1) are the average of 5 experimental results. And for all the EL tasks, since BIO tagging cannot address the problem, SPADE decoder is applied to all models.

### C.1 FURTHER ABLATION STUDIES ON ALL DOWNSTREAM TASKS

Table 6 and Table 7 are the extension of Table 4 and show the F1 scores for all downstream EE and EL tasks measured by changing each components one by one in BROS. From these tables, we can see that the settings of BROS show the best performance in most cases.

| Model | FUNSD EE | SROIE EE | CORD EE |
|---|---|---|---|
| BROS | 74.44 | 93.99 | 95.15 |
| (area-masking → token-masking) | 73.85 | 93.07 | 94.89 |
| (sinusoid & linear → look-up table) | 67.99 | 92.22 | 93.53 |
| (untied → tied) | 42.46 | 26.39 | 75.21 |
| (SPADE decoder → BIO tagging) | 73.16 | 93.70 | 94.88 |

Table 6: Ablation studies on all downstream EE tasks. For each ablation, the other components are set as those of BROS except for the highlighted component.

| Model | FUNSD EL | CORD EL | SciTSR EL |
|---|---|---|---|
| BROS (with SPADE decoder) | 42.48 | 90.59 | 99.17 |
| (area-masking → token-masking) | 41.92 | 89.80 | 99.15 |
| (sinusoid & linear → look-up table) | 32.11 | 86.55 | 98.65 |
| (untied → tied) | 27.83 | 76.35 | 98.76 |

Table 7: Ablation studies on all downstream EL tasks. For each ablation, the other components are set as those of BROS except for the highlighted component.

### C.2 GRADUALLY ADDING PROPOSED COMPONENTS TO THE ORIGINAL LAYOUTLM

To evaluate performance improvements from LayoutLM, we provide the experimental results when gradually adding each new component. Table 8 and Table 9 provide performance changes of F1 score for EE and EL tasks, respectively. In most cases, our proposed methods show performance improvements over all tasks.

| Model | FUNSD EE | SROIE EE | CORD EE |
|---|---|---|---|
| **LayoutLM** | 64.54 | 93.47 | 92.81 |
| + area-masking strategy | 65.98 | 92.97 | 93.56 |
| + untied token encoding | 66.78 | 93.70 | 93.46 |
| + positional embedding (sinusoid & linear) | 73.16 | 93.70 | 94.88 |
| + SPADE decoder (→ **BROS**) | 74.44 | 93.99 | 95.15 |

Table 8: Performance improvements on EE tasks through gradual changes from Layout's method to our method. At the last line, all components are changed from LayoutLM and the model becomes BROS.

| Model | FUNSD EL | CORD EL | SciTSR EL |
|---|---|---|---|
| **LayoutLM** (with SPADE decoder) | 29.18 | 86.29 | 98.35 |
| + area-masking strategy | 31.74 | 86.47 | 98.67 |
| + untied token encoding | 32.11 | 86.55 | 98.65 |
| + sinusoid & linear ($\rightarrow$ **BROS**) | 42.48 | 90.59 | 99.17 |

Table 9: Performance improvements on EL tasks through gradual changes from Layout's method to our method. At the last line, all components are changed from LayoutLM and the model becomes BROS.

### C.3    PROPOSED COMPONENTS ON THE ORIGINAL LAYOUTLM

For apples-to-apples comparison, we provides performance changes when adding each proposed component on LayoutLM. The results are shown in Table 10 and Table 11. When changing the original module to ours, the performances are solely increased except for the case of the positional embedding (sinusoid & linear). Interestingly, when combining our positional embedding and encoding (untied), the performance is dramatically increased. This result shows the benefits of using our proposed embedding and encoding methods together.

| Model | FUNSD EE | SROIE EE | CORD EE |
|---|---|---|---|
| LayoutLM | 64.54 | 93.47 | 92.81 |
| (token-masking $\rightarrow$ area-masking) | 65.98 | 92.97 | 93.56 |
| (tied $\rightarrow$ untied) | 66.03 | 94.23 | 93.38 |
| (look-up table $\rightarrow$ sinusoid & linear) | 49.42 | 79.16 | 87.76 |
| (tied + look-up table $\rightarrow$ untied + sinusoid & linear) | 72.48 | 93.61 | 94.89 |
| (BIO tagging $\rightarrow$ SPADE decoder) | 65.91 | 92.18 | 92.93 |

Table 10: Performance comparison on EE tasks when changing from Layout's method to our method. At each ablation, the other components are set as those of LayoutLM.

| Model | FUNSD EL | CORD EL | SciTSR EL |
|---|---|---|---|
| LayoutLM (with SPADE decoder) | 29.18 | 86.29 | 98.35 |
| (token-masking $\rightarrow$ area-masking) | 31.74 | 86.47 | 98.67 |
| (tied $\rightarrow$ untied) | 32.79 | 86.60 | 98.62 |
| (look-up table $\rightarrow$ sinusoid & linear) | 32.31 | 80.06 | 98.91 |
| (tied + look-up table $\rightarrow$ untied + sinusoid & linear) | 41.92 | 89.80 | 99.15 |

Table 11: Performance comparison on EL tasks when changing from Layout's method to our method. At each ablation, the other components are set as those of LayoutLM.

## D    RESOURCE ANALYSIS

Table 12 shows the resource and speed analysis of LayoutLM and BROS. The F1 scores of LayoutLM are referred from (Xu et al., 2020) and all pre-training models are trained with 1 epoch of 11M data. As can be seen, BROS shows better performance than LayoutLM$_{LARGE}$ even though requiring fewer parameters and less inference time.

| Model | Backbone | # params | inference time (ms) | FUNSD EE |
|---|---|---|---|---|
| LayoutLM$_{BASE}$ | BERT$_{BASE}$ | 113M | 100.46 | 76.36 |
| BROS | BERT$_{BASE}$ | 139M | 135.36 | 79.63 |
| LayoutLM$_{LARGE}$ | BERT$_{LARGE}$ | 343M | 292.08 | 77.89 |

Table 12: Resource analysis of LayoutLM and BROS. To measure the inference time, we used 1 NVIDIA Tesla V100 32GB GPU with a batch size of 8.

## E    GRAPH-BASED FORMALIZATION FOR EE AND EL TASKS

Document KIE is a task that extracts structural information from documents. In this paper, we defined EE task that identifies text sequences for target classes and EL task that links the head of the text sequences to determine structural information of documents. These EE and EL tasks can be interpreted as tasks that identify a directional graph structure between OCR text blocks. In this formalization, all tokens are treated as nodes in a graph and the links between the nodes indicate the structural relationships between tokens in a document. Figure 5 shows examples of FUNSD, SROIE, CORE, and SciTSR with the graph-based formalization.

## F    SAMPLE INFERENCE RESULTS OF THE FUNSD EE AND EL TASKS

Figure 6 shows the inference results of LayoutLM and BROS and the ground truth of a same FUNSD image. Even though the document has a complex layout, BROS identified key contexts and relations reasonably. However, we observed that LayoutLM tends to link unrelated contexts that are spatially far in the layout.

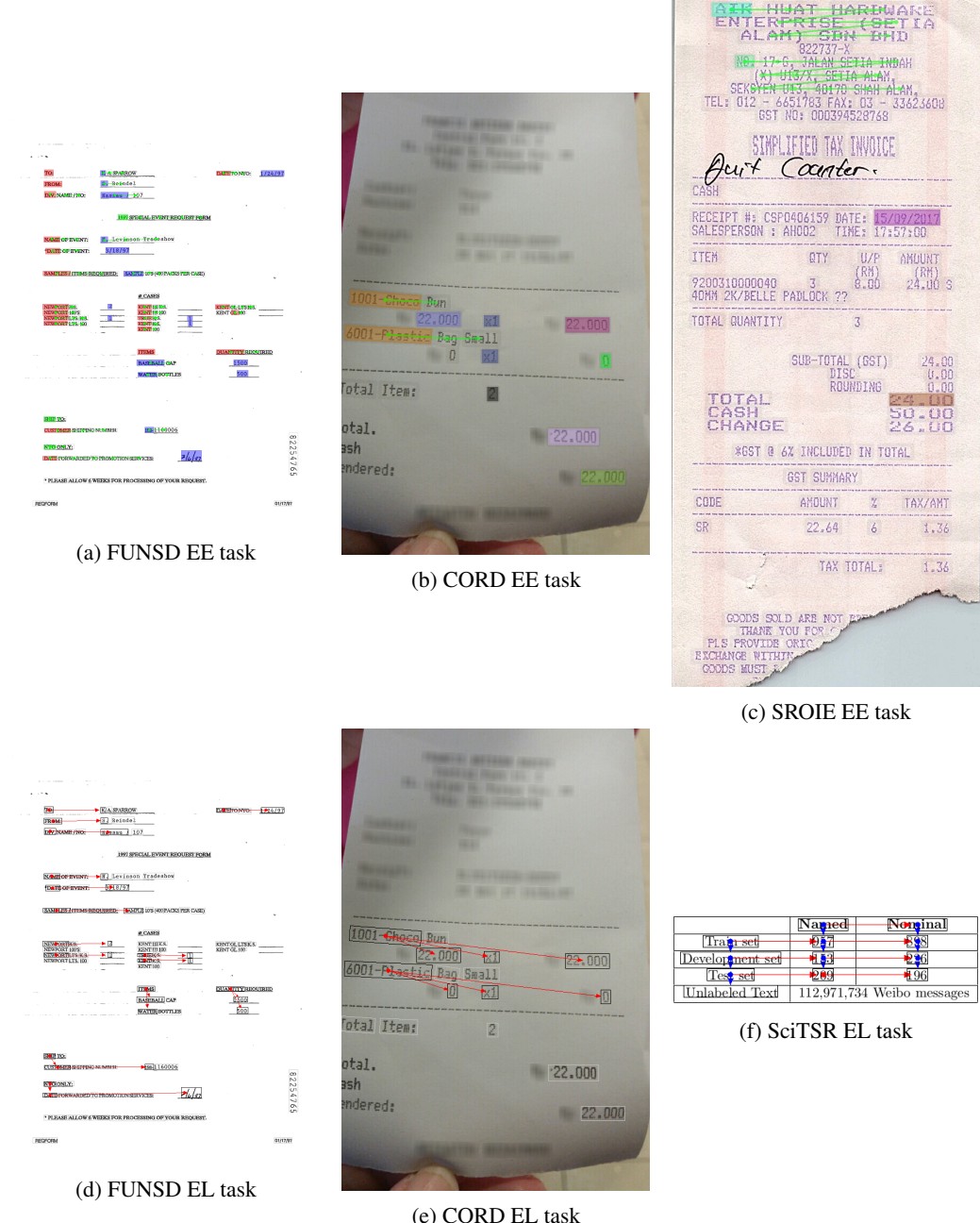

Figure 5: Examples of EE and EL task definition through graph-based formalization in all downstream tasks. For all EE tasks, the colored boxes indicate the first boxes of target classes and the green arrows represent next word connections. For example, the first menu in (b) has the first word is "1001-Choco" and its next word is "Bun". For EL tasks, the structural relationship between components are represented with red arrows. For example, the first menu ("1001-Choco Bun") in (b) has multiple sub information such as unit price ("22.000"), count ("x1"), and total price ("22.000"). SciTSR has two types of relations such as "horizontal" (red) and "vertical" (blue). As shown in the examples, EE and EL tasks can be formed with these graph-based formalization.

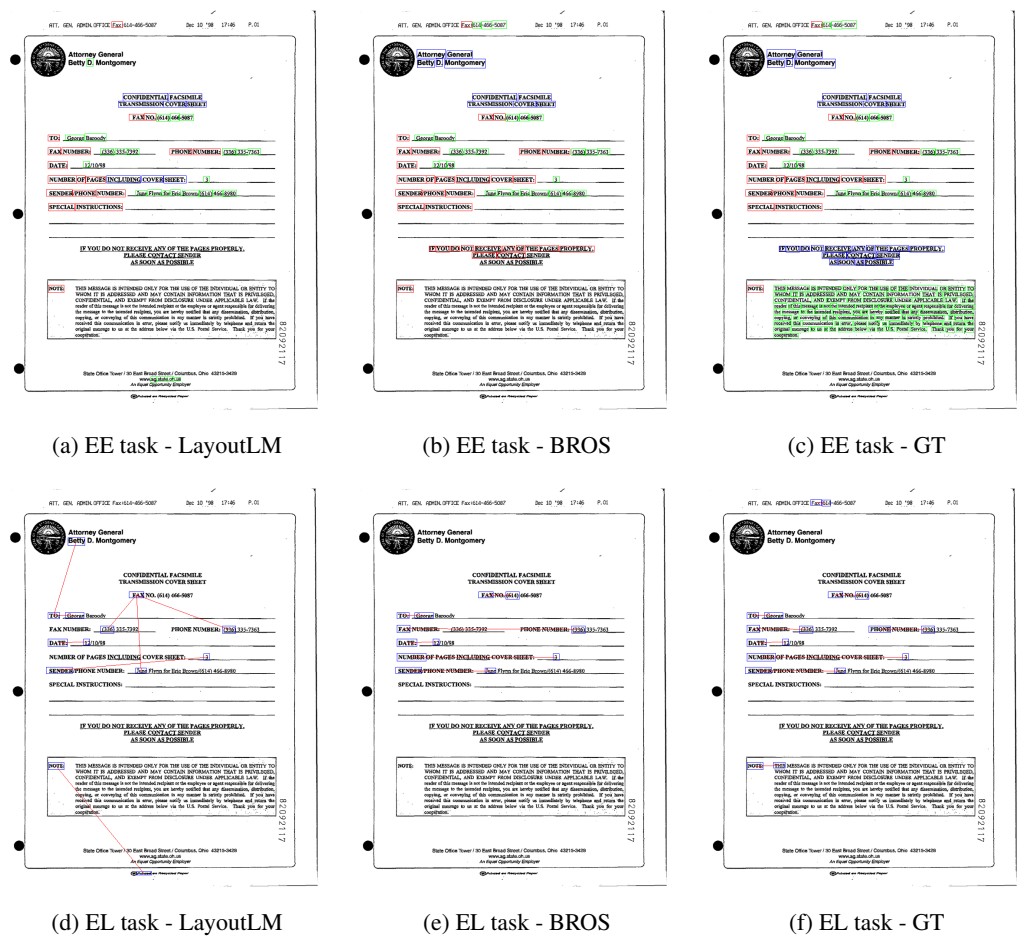

(a) EE task - LayoutLM     (b) EE task - BROS     (c) EE task - GT

(d) EL task - LayoutLM     (e) EL task - BROS     (f) EL task - GT

Figure 6: Inference EE and EL results on a sample FUNSD image. The images of upper row show the results of LayoutLM, BROS, and ground truth for EE task and the images of lower row show those for EL task. In EE task, the red, green, and blue boxes indicate Question, Answer, ans Header classes, respectively. In EL task, the blue boxes indicate key blocks and the red lines show the relations.

