# OpenReview forum: "BROS: A Pre-trained Language Model for Understanding Texts in Document"
_ICLR.cc/2021/Conference — Reject_

### Official Review · AnonReviewer1 · 2020-10-26
**Recommendation to Accept**

**Rating:** 6
**Confidence:** 3

**Review:**

Summary:

The paper provides a novel pretrained language model for document understanding named BROS, which adds spatial layout information and new area-masking strategy.
The authors do some experiments on four public datasets to illustrate the effectiveness of BROS.
The new architecture is well-suited for understanding texts in document, which is valuable.

##########################################################################

Reasons for score:


Overall, I vote for accepting.
I deem that the pre-trained language model based on BERT that encodes spatial information is useful for 2D document.
Hopefully the authors can address my concern in the rebuttal period (see cons below).


##########################################################################Pros:


1. The paper addresses siome limitations which are very important for document understanding: spatial information, spatial
relation, and the information of text blocks.


2.  This paper provides comprehensive experiments, including both qualitative analysis and quantitative results, to show the effectiveness of the proposed model.  The entire structure is organized well and the formulas are very detailed.


##########################################################################

Cons:

1. What are the advantages of BROS in terms of speed and resource consumption?
It would be more convincing if the authors can provide more cases in the rebuttal period.

2. For the Figure 1(b), it would be better to provide more details about it, which seems not very clear to me. Like how to
mask in red area?



##########################################################################

Questions during rebuttal period:


Please address and clarify the cons above


#########################################################################

---

> ### Author Response · Authors · 2020-11-13
> **Response to reviewer1**
>
> Thank you for your constructive comments. We are glad that you have agreed with the current limitations and our comprehensive experiments on them.
>
> > #### **A. Required resources**
>
> Model | backbone      | # params    | inference-time (ms) | FUNSD F1
> --- | --- | --- | ---  | ---
> LayoutLM_BASE | BERT_BASE    | 113M           | 100.46                         | 76.36
> BROS | BERT_BASE    | 139M           | 135.36                         | **79.63**
> LayoutLM_LARGE | BERT_LARGE | 343M           | 292.08                         | 77.89
>
> The above table shows the resource analysis of LayoutLM and BROS. The F1 scores of LayoutLM are referred from Xu et al. (2020) and all pre-training models are trained with 1 epoch of 11M data. As can be seen, BROS shows better performance than LayoutLM_LARGE even though requiring fewer parameters and less inference time. We will add this analysis in this discussion period.
>
> > #### **B. More description about Figure 1(b)**
>
> We found that the description of Figure 1 should be improved. In the figure, red boxes indicate random selections of text blocks/tokens and the black boxes represent the masked tokens. Although there can be multiple tokens in a single text block, the token masking (Figure 1.a) flats all tokens of a document, selects a token randomly (red), and masks it directly (black). In contrast, the area masking (Figure 1.b) selects a text block, identifies an area by expanding the block area with an exponential distribution (red), masks tokens in all text blocks of which center is aligned in the identified area (black). Thanks to this comment, we will improve the figure description.

---

> > ### Author Response · Authors · 2020-11-19
> > **Following response**
> >
> > Now, we updated the paper with improved description for Figure 1.(b) (Section 3.2) and resource analysis (Appendix D).

---

### Official Review · AnonReviewer4 · 2020-10-28

**Rating:** 5
**Confidence:** 3

**Review:**

### Overall

Authors used BERT alongside to a 2D-position embedding based on a sinusoidal function and a graph-based decoder to improve performance on document information extraction tasks. They do pre-train their model (BROS) on a large dataset with 11M documents, and then used such models to perform downstream tasks in four smaller datasets. Their models achieve better quantitative results when compared to the provided baselines.

### Positive aspects

* Positional encoder based on sinusoidal function seems to be effective.
* Authors perform experimentally sound experiments, following closely LayoutLM.
* Pre-trained models could be useful.
* Authors reproduced results from their strongest baseline.
* Better results in all downstream tasks.

### Cons and aspects to improve

My main concern is that the overall contribution is seems to be limited.In fact, the original paper of the Transformer approach, already proposed such kind of embedding. It is good to know that it works for 2D-coordinates for the task at hand, though it seems to be more a marginal improvement on existing work rather than a standalone contribution.

It is hard to tell what are the standalone contributions of the paper, and what is coming from other works.

Authors could have provided more in-depth details (visualizations, analysis, examples) to show main differences between the proposed approach and baselines (specially LayoutLM). Also they could visually demonstrate the advantages of their approach.

Authors could have plugged their embedding strategy in LayoutLM to understand the impact of that particular component.

I would like to have seen qualitative examples of model predictions, and more examples from the dataset.

A figure containing the whole process could be helpful to better understand the processing required to train / test such models. Figure from LayoutLM is a good example of that, it comprises the entire process and makes it easier to understand the whole architecture.

* In the abstract, authors say "BROS utilizes a powerful graph-based decoder that can capture the relation between text segment"* Though in the text such a component (that is from other work) is only mentioned twice without further detail.

It is unclear to me:

* How regions of interest are detected in this work? (I assumed authors used the same strategy as LayoutLM).
* OCR seems to be an extra-step in the preprocessing stage. What to do if the user does not have the same OCR. What is the impact of a good OCR for training and testing (prediction of new, unseen documents)?
* "In-house OCR engine was applied" can authors provide more details on that?

This line of work could be much stronger if the models comprised the whole process (detection, text extraction, recognition) in an end-to-end manner.

### Notes on text and style

There are parts of the manuscript that felt somewhat informal and confusing to me. I will provide some details as follows.

* Set a default format for numbers in tables. Table 1 has two distinct decimal number formats.
* Personally, I think it is better to write $5 \times 10^{-5}$ rather than 5e-5.
* In the results section there is a typo: *"performances with a large margins of 2.32pp in"*. Also, text could be more formal. I would avoid using the use of the *pp* abbreviation.
* "By achieving the best, these results prove that BROS" this sentence can be improved.
* "Moreover, it should be noted that BROS achieves higher f1 score than 79.27 of LayoutLM using visual features". I think authors wanted to say that even though BROS does not rely on visual features, it does outperform LayoutLM which, in turn, uses visual features.

---

> ### Author Response · Authors · 2020-11-13
> **Response to AnonReviewer4 (2/2)**
>
> > #### **F. Discussion about the end-to-end model (from raw image to key texts)**
>
> Developing an end-to-end model is an interesting and desirable research topic. Recently, in table detection and recognition, TableNet(Paliwal et al., 2019), DeepDeSRT(Schreiber et al., 2017), and CascadeTabNet(Prasad et al., 2020) have been introduced as end-to-end models including the functionality of OCR. We hope that an end-to-end model for document KIE tasks will be introduced in the future and expect that BROS will be a stepping stone for the goal.
>
> *Paliwal et al., TableNet: Deep Learning model for end-to-end Table detection and Tabular data extraction from Scanned Document Images, ICDAR-19.*
>
> *Schreiber et al., DeepDeSRT: Deep learning for detection and structure recognition of tables in document images, ICDAR-17.*
>
> *Prasad et al., CascadeTabNet: An approach for end-to-end table detection and structure recognition from image-based documents, CVPR Workshop-20.*
>
> > #### **G. Pointed texts and styles that can be improved**
>
> Thank you for pointing out the sentences that can be improved. As you suggested, we will fix the sentences.

---

> > ### Author Response · Authors · 2020-11-19
> > **Following response**
> >
> > We added examples for each downstream tasks (See Appendix E), visual description of BROS (See Section 3), and visual comparison from the baseline (See Appendix F). We hope that our responses and these updates help to answer your concerns.
> >
> > We hope to keep constructive discussion. If there is something missing, please let us know.

---

> ### Author Response · Authors · 2020-11-13
> **Response to AnonReviewer4 (1/2)**
>
> Thank you for your contributions. Followings are discussions about your major and minor concerns.
>
>  > #### **A. Not a standalone contribution**
>
> BROS tackles document KIE tasks by addressing current limitations on multiple components. They are standalone contributions: (1) area-masking for learning text spans on 2D space, (2) positional embedding and encoding that have not been applied for 2D language models, (3) the combination between pre-trained model and SPADE decoder for document KIE tasks, and (4) evaluation on realistic scenarios without perfect serialization. Following the "No Free Lunch Rule", each function and metric might not be new but it should be introduced in this research area. In fact, several pre-trained language models such as BERT, RoBERTa, SpanBERT, StructBERT contribute to the NLP field by optimizing or improving the learning process without any architectural changes. As like them, this paper also contributes document intelligence by providing a pre-trained 2D language model improved from LayoutLM in multiple aspects.
>
>  > #### **B. Ablation study based on LayoutLM**
>
> Model| FUNSD EE F1
> --- | ---
> LayoutLM | 64.54
> (with our learning strategy)| 65.98
> (with our position embedding)| 49.42
> (with our position encoding)  | 66.03
> (with our position embedding \& encoding) | 72.48
> (with our decoder) | 65.91
>
> Thank you for your constructive suggestion. As you requested, we provide experiment results by plugging the proposed modules into LayoutLM. When changing the original module to ours, the performances are solely increased except for the case of the positional embedding (sinusoid \& linear). Interestingly, when combining our positional embedding and encoding (untied), the performance is dramatically increased. This result shows the benefits of using our proposed embedding and encoding methods together. We will add this ablation study in the paper.
>
>  > #### **C. More examples from the dataset**
>
> We will add more examples of the datasets and figures comparing the results of LayoutLM and BROS.
>
>  > #### **D. Model overview figure**
>
> We will add the overview figure of BROS for a better description.
>
>  > #### **E. Questions**
>
> Q1) How regions of interest are detected in this work? (I assumed authors used the same strategy as LayoutLM).
>
> We guess that the region of interest will be red boxes in Figure 1(b). Our approach randomly selects a text block and expands the text region with exponential distribution. The intuition behind this approach is that area covering text spans on 2D space should be related to the size and location of the texts.
>
> Q2) OCR seems to be an extra-step in the preprocessing stage. What to do if the user does not have the same OCR. What is the impact of a good OCR for training and testing (prediction of new, unseen documents)?
>
> We provide two answers: for pre-training and downstream tasks. If we didn't figure out your question, please reply to discuss this correctly.
>
> * Pre-training) As you recognized, IIT-CDIP (11M) is pre-processed and we pre-trained LayoutLM$^{\dagger}$ and BROS on the same dataset. Interestingly, LayoutLM$^{\dagger}$ performs similarly with the original LayoutLM pre-trained on their OCR results in multiple settings (See Appendix A.1). In other words, two advanced OCR engines reach similar results when applied to the dataset for pre-training. It is over-burden to prove the effect of OCR engines, but the recent OCR engines will be promising for pre-training 2D language models.
>
> * Downstream) To train and evaluate downstream tasks with their ground truth annotation, we utilize the provided OCR results without applying other OCR engines. However, the labels of downstream tasks are tied with provided text blocks and the labels cannot be defined on the other OCR results.
>
> Q3) "In-house OCR engine was applied" can authors provide more details on that?
>
> The OCR engine consists of a CRAFT detector (Baek et al., 2019) and an ASTER recognizer (Shi et al., 2018). To keep anonymity, it is hard to describe more details but we will provide the API information that we used after the discussion period ends.
>
> *Baek et al., Character Region Awareness for Text Detection, CVPR-19.*
> *Shi et al., ASTER: An Attentional Scene Text Recognizer with Flexible Rectification, TPAMI-18.*

---

### Official Review · AnonReviewer2 · 2020-10-28
**This paper largely overlaps with pervious research work with minor modifications**

**Rating:** 5
**Confidence:** 5

**Review:**

The paper proposes the pre-trained language model BROS which aims to leverage both text and spatial information to improve information extraction on documents. Using the graph-based decoder from SPADE, BROS achieves the SOTA performance on some entity extraction and entity linking downstream tasks. However, the area-masking strategy does not show significant improvement over the LayoutLM and the graph decoder is proposed in SPADE which is not new. In addition, as most commercial OCR tools have already got very good reading order information, the benefit from the graph decoder might be marginal.

Pros
-	The paper introduces the area-masking pre-training strategy that can be seen as a natural generalization of masking language model in the 2D plane.
-	The authors integrate spatial information into the attention mechanism as a pair-wise bias term, which is reasonable.
-	BROS utilizes the graph-based decoder from SPADE and improves performance on downstream tasks.

Cons
-	The area-masking strategy is to mask small area centered at some tokens, which is actually similar to masking the center token only. Also, given that the FUNSD dataset is small, the area-masking strategy does not show significant improvement over vanilla MLM.
-	This paper shows that sinusoid & linear functions can encode 2D position efficiently. However, it is not reasonable to compare sinusoid & linear and learnable embeddings on small data, since learnable embeddings could leverage large amount of data and get more gains.
-	The graph-based decoder part is identical to which in SPADE so it is not suitable to appear as the contributions of this paper.

In summary, this paper largely overlaps with the previous research work. I do not think it is qualified for the ICLR conference.

---

> ### Author Response · Authors · 2020-11-13
> **Response to AnonReviewer2**
>
> Thank you for your services. We found that your concerns lied on the area-masking strategy and the graph decoder. However, we argue that our masking strategy is distinct from the previous approach as well as that the graph decoder is necessary for a real scenario. The followings provide detailed discussions about your concerns.
>
> > #### **A. More description about the area-masking strategies**
>
> To make an area hide text span (n-gram) in 2D space, the area should be related to the text sizes and locations in a document. Our approach expands an area of a randomly selected text block with an exponential distribution. After an area is identified, all text blocks are evaluated whether their centers are allocated in the area or not, then all tokens in the allocated blocks are masked. Such masking process is repeatedly conducted until the number of masked tokens reaches to 15\% of the total. As you mentioned, we found that the description of Figure 1 should be improved and we will add more descriptions in the paper.
>
> > #### **B. Marginal performance gain from the area-masking strategy**
>
> As you mentioned, the improvement from the area masking might look marginal in the small FUNSD dataset. When we conducted the ablation study on other datasets, we observed consistent improvements over all EE downstream tasks. See the following table.
>
> MLM | FUNSD EE | SROIE EE | CORD EE |
> --- | --- | --- | --- |
> Random tokens | 73.85 (0.40) | 93.07 (0.52) | 94.89 (0.24) |
> Random areas   | 74.44 (0.36) | 93.99 (0.43) | 95.15 (0.22) |
>
> (Numbers: average and standard deviation of five F1 scores)
>
> > #### **C. Comparison between sinusoid \& linear and learnable embeddings for representing 2D positions**
>
> Embedding | FUNSD EE |
> --- | --- |
> Look-up table | 78.81 |
> Sinusoid \& linear | 79.63 |
>
> Your concern was that the comparison of the embedding methods was conducted with the models pre-trained only with 512K data.
> We agreed with your concern, so we conducted a comparison on the cases with 11M data (11M, 1 epoch).
> The results show that our embedding is efficient as well as effective when compared to LayoutLMs'. We will provide more comparisons on other downstream tasks.
>
> > #### **D. Difficulty on serializing OCR blocks**
>
> We don't agree with your opinion: "as most commercial OCR tools have already got very good reading order information, the benefit from the graph decoder might be marginal.". First, serializing text blocks is emerging research problem (Li et al., 2020). Second, the most commercial OCR tools might look good but they fail to organize a layout document in a reading order (See Appendix B). Finally, EL tasks from FUNSD, CORD, and SciTSR require the graph decoder because BIO tagging approach cannot address the problem. It should be noted that the graph decoder is applied to all models including LayoutLM for EL tasks.
>
> In addition, even though holding perfect serialization information, the graph decoder shows consistent improvement from the BIO tagger. See the following table.
>
> Decoder | FUNSD EE | SROIE EE | CORD EE |
> ---| --- | --- | --- |
> BIO tagger | 73.16 | 93.70 | 94.88 |
> SPADE decoder| 74.44 | 93.99 | 95.15 |
>
> We will add this table and analysis on the paper. It will be constructive to discuss this issue more if we misunderstood your opinion.
>
> *Li et al., An End-to-End OCR Text Re-organization Sequence Learning for Rich-text Detail Image Comprehension, ECCV-20.*
>
> > #### **E. Difference from SPADE**
>
> As you mentioned, proposing SPADE decoder is not our contribution so we have not argued that in our paper. SPADE (Hwang et al., 2020) is a transformer-based model including a graph-based decoder. Although the graph-based formalism requires spatial relationships between text blocks, they suffered from learning them only with small datasets (downstream datasets in our paper). To compensate for the issue, BROS learns spatial encoding with large-scale data and fine-tunes that to support the SPADE decoder. Our paper proved that the SPADE decoder with BROS was suitable for document KIE tasks rather than argued that SPADE itself was our main contribution.

---

> > ### Author Response · Authors · 2020-11-19
> > **Following response**
> >
> > From your valuable comments, the improved paper is now available. The description for area-masking is improved (Section 3.2) and the ablation studies on multiple tasks are added (Appendix C.1).
> >
> > We hope to keep this constructive discussion. If there is something missing, please let us know.
> >
> >
> > > #### **C. Comparison between sinusoid & linear and learnable embeddings for representing 2D positions (More comparison on other datasets)**
> >
> > As we mentioned before, we provide more results (11M, 1 epoch) comparing sinusoid & linear and learnable embeddings on other downstream tasks.
> >
> > Embedding       | FUNSD EE | SROIE EE | CORD EE | FUNSD EL | CORD EL | SciTSR EL |
> > --- | --- | --- | --- | --- | --- | ---
> > Look-up table   | 78.81 | 95.15 | 95.32 | 55.04 | 91.00 | 99.09
> > Sinusoid \& linear | 79.63 | 94.87 | 95.46 | 63.83 | 91.98 | 99.29
> >
> > As can be seen, "Sinusoid \& linear" looks better in the case of pre-training large-scale documents (11M). These experiments are conducted with our untied positional encoding that showed promising improvement compared to the tied positional encoding. We hope that this result can respond to your concerns.

---

### Official Review · AnonReviewer3 · 2020-10-29
**A robust and effective pre-training strategy for document understanding and is independent of optimal order information but lacks of some details**

**Rating:** 6
**Confidence:** 4

**Review:**

> Summary:

The paper studies the problem of large-scale pre-training for semi-structured documents. It proposes a new pre-training strategy called BERT relying on Spatiality (BROS) with area-masking and utilizes a graph-based decoder to capture the semantic relation between text blocks to alleviate the serialization problem of LayoutLM.

It points out that LayoutLM fails to fully utilize spatial information of text blocks and will face difficulties when text blocks cannot be easily serialized.

The three drawbacks of LayoutLM are listed:
* X-axis and Y-axis are treated individually with point-specific embedding
* Pre-training is identical to BERT so does not consider spatial relations between text blocks
* Suffer from the serialization problem

The proposed three corresponding methods of BROS are:
* Continuous 2D positional encoding
* Area-masking pre-training on 2D language modeling
* Graph-based decoder for solving EE & EL tasks

> Strength:

* The paper makes incremental advances over past work (LayoutLM) and the proposed BROS models achieves SOTA performance on four EE/EL datasets (i.e., FUNSD, SORIE*, CORD, and SciTSR)

* The paper is generally easy to follow and could be better if provide more important details in Section 3.2 & 3.3

* The experiment and discussion for Section 5.3 are quite convincing. BROS could achieve robust and consistent performances across all the four permuted version datasets, which demonstrates that BROS is adaptive to documents from the practical scenarios.

> Major concerns:

- For Section 3.2, the author didn’t even provide the pre-training objective for the area-masked language model. I think the author should include the objective and define the exponential distribution explicitly.

* I’m confused about why the performance difference in Table 4 between original LayoutLM and BROS is larger than that in Table 1. In the original LayoutLM, the 2D position encoding method is tied with token embedding. This applies to both Table 1 and Table 4. However, in Table 4 the performance difference on FUNSD EE is 42.46 vs 74.44, while in Table 1 the performance comparison is 78.89 vs 81.21. Could the author give some explanations on this?

- In Table 4, it is better for the author to clearly indicate each ablation’s components. The author should also give the performance of the original LayoutLM and performances after gradually adding each new component to the original LayoutLM: such as original LayoutLM + Sinusoid & Linear, original LayoutLM + Sinusoid & Linear + untied with token embedding, etc.

* For encoder design in Eq. (2), the second term is used to model the spatial dependency given the source semantic representation.  How about adding a symmetric term to model the semantic dependency given the source spatial representation. My question is simply that why not adopt a symmetric design for Eq. (2)?

* Can the author give the reason behind the design of $\mathbf{t}^{ntc}$ in Eq.(4)? I’m not so clear about the function of $\mathbf{t}^{ntc}$.  Does the $\mathbf{f}_{ntc}$ output a distribution of the probability to be the next token over all N tokens?

* Could the author give a detailed analysis on which strategy contributes the most to BROS’ robustness against permuted order information? From the results of Table 4, it is not the SPADE decoder and the most important factor seems to be calculating semantic and position attentions separately, i.e., untied with token embedding and explicitly modeling semantic/position relations between text blocks. Do the authors agree with my conjecture?

> Minor concerns:

* Although SPADE is not the most important component of BROS, I believe including details of the grade-based decoder will help the readers to understand the model much better.

* I’m curious about the performance of SpanBERT on the four datasets since the author said that area-masking of BROS is inspired by SpanBERT.

* In Table 3, the value of LayoutLM - FUNSD should be 78.89 since all other p-FUNSD & FUNSD related values are consistent with Table 1 & 2.

---

> ### Author Response · Authors · 2020-11-13
> **Response to AnonReviewer3**
>
> Thank you for your detailed and constructive comments on our work.
>
> > #### **A. Pre-training objective for area-masking LM**
>
> We will add the pre-training objective with a mathematical formula in the paper. Thank you for the comment.
>
> > #### **B. Performance difference between Table 1 and Table 4**
>
> The performances in Table 1 and Table 4 cannot be simply compared. First, in Table 4, we set the basic setting as BROS and change our proposed modules to the original modules (LayoutLMs') to prove that all components are important to improve performances. Second, in Table 4, the models are trained with a subset of the training dataset (512K documents) to conduct multiple experiments with limited resources. Since pre-training with a large-scale dataset takes time, their large-scale experiment cannot be provided in this discussion period. However, if you want, we are willing to provide the results at the camera-ready version.
>
> > #### **C. Ablation study: gradually adding each new component to the original LayoutLM**
>
> Thank you for your constructive suggestion. The following table provides the performances when gradually adding each new component. As can be seen, the performance is also improved gradually. We will add this experiment to the paper.
>
> Model | FUNSD EE | SROIE EE | CORD EE
> --- | --- | --- | ---
> LayoutLM                | 64.54 | 93.47 | 92.81
> +area-masking          | 65.98 | 92.97 | 93.56
> +Untied with token emb | 66.78 | 93.70 | 93.46
> +sinusoid \& linear    | 73.16 | 93.70 | 94.88
> +SPADE (BROS)          | 74.44 | 93.99 | 95.15
>
> Model (all SPADE) | FUNSD EL     | CORD EL      | SciTSR EL
> --- | --- | --- | ---
> LayoutLM                    | 27.06 | 87.71 | 98.35
> +area-masking              | 28.19 | 88.12 | 98.67
> +Untied with token emb     | 26.95 | 88.03 | 98.65
> +sinusoid \& linear (BROS) | 40.04 | 91.00 | 99.17
>
> > #### **D. Symmetric design for calculating attention score**
>
> The motivation of the second term in Eq.(2) is to enable the model to find the target position effectively when the query context and position are given. For example, in the case of Figure 2, the context and position ("Date") requests the position of ("December 9, 1999"). In our early experiment, the symmetric approach that you mentioned showed performance degradation. In addition, when utilizing SPADE, we observed that the degree of degradation increased. We think that the opposite direction of the conditional attention score might be harmful to identify spatial relationships between text blocks.
>
> Model              | Decoder | FUNSD EE | SROIE EE | CORD EE |
> --- | --- | --- | --- | ---
> BROS (with eq.(2)) | BIO     | 73.16    | 93.70    | 94.88   |
> BROS (symmetric)   | BIO     | 71.58    | 93.78    | 94.01   |
> BROS (with eq.(2)) | SPADE   | 74.44    | 93.99    | 95.15   |
> BROS (symmetric)   | SPADE   | 62.70    | 82.20    | 90.02   |
>
> > #### **E. Design of $t^{\text{ntc}}$ in Eq.(4)**
>
> Thank you for pointing out the description about Eq.(4). As you mentioned, the output of $f_{\text{ntc}}$ is a distribution over N+1. Here, +1 indicates that the token does not have a next token or the token is not related to any class (similar role with an end-of-sequence [EOS] in NLP). $t^{\text{ntc}}$ is introduced to identify the un-connected tokens. We found that the description should be improved and we will provide a better description for Eq.(4).
>
> > #### **F. Which strategy contributes the most to BROS’ robustness against permuted order information**
>
> SPADE decoder is the most important factor for permuted datasets because BIO encoding cannot group text blocks without their perfect order information (SPADE decoder is used in Table 2 and 3). To clarify the effectiveness of the proposed components on the permuted dataset, we are implementing additional ablation studies and we will share the results.
>
> > #### **G. Minor concerns**
>
> G1. "Although SPADE is not the most important component of BROS, I believe including details of the grade-based decoder will help the readers to understand the model much better."
>
> Thank you for your suggestion. We will provide more details about SPADE decoder.
>
> G2. "I’m curious about the performance of SpanBERT on the four datasets since the author said that area-masking of BROS is inspired by SpanBERT."
>
> Thank you for the interesting question. We will conduct the downstream tasks with SpanBERT.
>
> G3. "In Table 3, the value of LayoutLM - FUNSD should be 78.89 since all other p-FUNSD \& FUNSD related values are consistent with Table 1 \& 2."
>
> The score in Table 1 and 3 cannot be related because SPADE decoder is utilized in all experiments in Section 5.3. In the reason, LayoutLM$^{\dagger}$ shows the same score in Table 2 and 3 but it has different scores in Table 1 (BIO) and 3 (SPADE). We will clarify this difference in the paper.

---

> > ### Author Response · Authors · 2020-11-19
> > **Following response**
> >
> > Thanks for your constructive comments, we add pre-training objective (Section 3.2), more description for Eq.4 (Section 3.3), more description for graph-based formalization (Appendix E), ablation studies (Appendix C.2). The followings are additional responses.
> >
> > > #### **F. Which component contributes the most to BROS’ robustness against permuted order information**
> >
> > As we mentioned above, BIO-tagging cannot address document KIE tasks when there is no order information of target key texts. For examples, if the order of the key texts is wrong, BIO formalization cannot find the correctly ordered texts. Here, we provide additional ablation study to describe how the positional embedding and encoding methods contribute to provide the robustness on FUNSD and p-FUNSD under the use of SPADE decoder.
> >
> > \# | 2D embedding | 2D encoding  | decoder                | FUNSD | p-FUNSD
> > --- | --- | --- | --- | --- | ---
> > BROS | Sinusoid\&linear | Untied | SPADE                  | 74.44 | 45.16
> >  | Look-up table (LayoutLM's) | Untied | SPADE            | 67.99 | 21.10
> >  | Sinusoid\&linear | Tied (LayoutLM's) | SPADE           | 42.46 | 38.77
> >  | Look-up table (LayoutLM's) | Tied (LayoutLM's) | SPADE | 68.08 | 20.84
> >
> > There were three interesting observations. First, the combination of Layout's components show the second best on FUNSD but the worst performance on p-FUNSD. It indicates that the combination of LayoutLM's embedding and encoding can be acceptable on KIE tasks with optimal order information but it has a limitation in a realistic scenario. Second, the encoding method is critical on FUNSD and the embedding method is effective on p-FUNSD. Finally, the combination of our embedding and encoding shows the best performance compared to the other combinations.
> >
> > > #### **G.2. SpanBERT experiments**
> >
> > As you suggested, we fine-tuned SpanBERT for our document KIE tasks. Interestingly, SpanBERT shows better performance than BERT. We guess that SpanBERT learns long-term dependencies for text spans, so it extracts key texts and their relationship better.
> >
> > Model | FUNSD EE | FUNSD EL
> > --- | --- | ---
> > BERT | 60.26 | 0.98
> > SpanBERT | 63.12 | 12.65
> > BROS | 81.21 | 66.96

---

### Author Response · Authors · 2020-11-13
**Response to all reviewers**

Thank all reviewers for the constructive comments and the service to ICLR-21. For live discussion, we decided to respond to the comments first, then update the paper later in this discussion period. We will notice the improved points of the paper after updates.

---

> ### Author Response · Authors · 2020-11-19
> **The updated paper is now available.**
>
> We thank all reviewers for their contributions. Now, we updated the paper that is improved with all reviewers' valuable comments. The following lists the changes from previous versions.
>
> - Overview figure in Section 3 (from AnonReviewer 4)
> - Improved description for area-masking in Section 3.2 (from AnonReviewer 1, 2, and 3)
> - Improved description for graph-based formalization in Section 3.3 and Appendix E (from AnonReviewer3)
> - More ablation studies and resource analysis in Appendix C.1, C.2, C.3, and D (from all reviewers)
> - Description for the difficulty of the serialization in Appendix B (from AnonReviewer 2)
> - More examples and visual comparison in Appendix F (AnonReviewer 4)
>
> For the discussion about the components, see our individual responses.

---

### Decision · Program_Chairs · 2021-01-07
**Final Decision**

**Decision:**

Reject

**Comment:**

The paper proposes a new pre-trained language model for information extraction on documents. It consists of a new pre-training strategy with area-masking and a new graph-based decoder to capture the relationships between text blocks. Experimental results show better performances of the proposed approach.

Pros • The paper is generally clearly written. • Experimental results show better performances on the benchmark datasets.

Cons • Novelty of the work might not be enough. For example, the graph-based decoder is not new. The masking technique is also a natural extension of that in BERT. • Significance of the work might not be enough. For example, the improvement from the area masking is not so significant. • There are additional experiments which can be added, as pointed out by Reviewer 3. • Presentation can be further improved. Some of the issues indicated by the reviewers have been addressed in the rebuttal. We appreciate the authors’ efforts.

During the rebuttal, the authors have addressed the clarity issues pointed out by the reviewers. However, the main issues in novelty and significance still exist. The reviewers think that the quality of the work is still not high enough as an ICLR paper.